# Associations between autistic traits and early ear and upper respiratory signs: a prospective observational study of the Avon Longitudinal Study of Parents and Children (ALSPAC) geographically defined childhood population

Amanda Hall [1], Richard Maw,[2] Yasmin Iles-Caven,[3] Steven Gregory,[3] Dheeraj Rai,[4] Jean Golding [3]

[1]Life and Health Sciences, Aston University, Birmingham, UK
[2]UBHT, Retired ENT Consultant, Bristol, UK
[3]Bristol Medical School (PHS), University of Bristol, Bristol, UK
[4]Centre for Academic Mental Health, Bristol Medical School, University of Bristol, Bristol, UK

**Correspondence to**
Dr Jean Golding;
jean.golding@bristol.ac.uk

## ABSTRACT

**Objective** To determine whether early ear and upper respiratory signs are associated with the development of high levels of autistic traits or diagnosed autism.

**Design** Longitudinal birth cohort: Avon Longitudinal Study of Parents and Children (ALSPAC).

**Setting** Area centred on the city of Bristol in Southwest England. Eligible pregnant women resident in the area with expected date of delivery between April 1991 and December 1992 inclusive.

**Participants** 10 000+ young children followed throughout their first 4 years. Their mothers completed three questionnaires between 18–42 months recording the frequency of nine different signs and symptoms relating to the upper respiratory system, as well as ear and hearing problems.

**Outcome measures** Primary—high levels of autism traits (social communication, coherent speech, sociability, and repetitive behaviour); secondary—diagnosed autism.

**Results** Early evidence of mouth breathing, snoring, pulling/poking ears, ears going red, hearing worse during a cold, and rarely listening were associated with high scores on each autism trait and with a diagnosis of autism. There was also evidence of associations of pus or sticky mucus discharge from ears, especially with autism and with poor coherent speech. Adjustment for 10 environmental characteristics made little difference to the results, and substantially more adjusted associations were at p<0.001 than expected by chance (41 observed; 0.01 expected). For example, for discharge of pus or sticky mucus from ears the adjusted odds ratio (aOR) for autism at 30 months was 3.29 (95% CI 1.85 to 5.86, p<0.001), and for impaired hearing during a cold the aOR was 2.18 (95% CI 1.43 to 3.31, p<0.001).

**Conclusions** Very young children exhibiting common ear and upper respiratory signs appear to have an increased risk of a subsequent diagnosis of autism or demonstrated high levels of autism traits. Results suggest the need for identification and management of ear, nose and throat conditions in autistic children and may provide possible indicators of causal mechanisms.

## STRENGTHS AND LIMITATIONS OF THIS STUDY

⇒ The strength of these results lies in the fact that the data were collected prospectively using a geographically defined population of young children.
⇒ Two of the autism traits were identified using well validated scales (social communication and speech coherence).
⇒ There was no bias in the way in which the questions were asked or in the later identification of autism or autism traits.
⇒ A limitation was that not all children in the study were assessed for a diagnosis of autism.

## INTRODUCTION

Autism spectrum disorder, henceforward named as 'autism', is a fairly uncommon, yet heterogeneous, condition.[1] Its aetiology is complex and likely to involve a combination of gene–environment–biology interactions, with no one single cause.[2] There is increasing recognition that the aetiology of each autistic trait may differ, either genetically, epigenetically and/or conditionally on an environmental exposure.[2]

Understanding and addressing early comorbidities of children diagnosed with autism may lead to longer term improvements in quality of life and also inform the common mechanisms and pathways in the origins of autism.[3] Ear, nose and throat (ENT) conditions, including related hearing disorders, have been implicated in the development of autism,[4 5] including acute otitis media (ear infections), otitis media with effusion (OME or glue ear), temporary conductive hearing loss associated with otitis media, and sleep disordered breathing. For example, there

have been a number of studies using population health records comparing the frequency of ear infections in children with autism compared with controls. Most showed greater odds of ear infections in children with autism: for example, a US population sample of 483 autism children with 84 789 controls found an odds ratio (OR) of 3.1 (95% CI 2.1 to 4.7) for at least three ear infections[6]; others have also shown excesses of ear infections.[7 8] Conversely, two studies using Kaiser Permanente health records[9] showed no excess of ear infections among children with autism, but one did find an increased odds (OR 1.93, 95 CI 1.56 to 2.38) of ENT conditions associated with autism compared with controls.[5] Other conditions that have been implicated include sleep-disordered breathing or obstructive sleep apnoea in the early life history of children with autism in the USA[5 10] and Japan.[11] Others have shown an increased likelihood of surgery for a sleep disorder, such as adenoidotonsillectomy.[12]

Of other ENT conditions, there are fewer studies examining the early history of chronic OME, which is more likely than acute otitis media to interfere with hearing, although less easy to identify from health records. One method is to examine the rate of insertion of ventilation tubes, although it is not always possible to distinguish ventilation tubes inserted for OME from insertion for acute ear infections. Nevertheless, ventilation tube rates have been shown to be higher in children with autism.[7 13 14] Case–control studies comparing auditory function in children with autism have found more abnormal tympanograms in children suggesting a higher prevalence of OME,[15] and more auditory abnormalities.[16] Although other studies did not find more auditory abnormalities in children with autism,[17 18] in both these studies children were excluded if they had a history of ear infections and middle ear effusion.

Overall, therefore, the evidence suggests an increased prevalence of ENT and related hearing conditions in children with autism compared with typically developing children. Much of the evidence cited above is from health records which may be biased due to greater levels of help-seeking from health services of parents of children with suspected autism. Where a parent has concerns about their child's general development, they may be more likely to attend health services, resulting in increased recording of the presence of common health conditions even where the underlying prevalence is no more common than in the general population.

Longitudinal population cohort studies avoid many of the biases in studies using health records, but there are few such studies in the literature apart from that of Jeans and colleagues,[19] who analysed data from the Early Childhood Longitudinal Study Birth Cohort and found an increased OR of 1.46 (95% CI 1.24 to 1.72) of early ear infections in children with a later autism diagnosis. The present study, positioned within the longitudinal birth cohort ALSPAC (Avon Longitudinal Study of Parents and Children), uses information collected prospectively from birth to assess whether any early indications of ENT symptoms,

including ear infections and intermittent hearing problems, are associated with one or more of the traits shown to be strongly related to a diagnosis of autism. Its aim is to investigate whether children with autism, and those with high levels of autistic traits or a diagnosis of autism, were recorded as having more early ear and upper respiratory signs and symptoms than expected by chance.

## METHODS

### Study population

ALSPAC is a pre-birth cohort study that enrolled ~80% of pregnant women resident in the Avon area of the UK in 1991–2. The initial number of pregnancies enrolled was 14 541 (for these at least one questionnaire had been returned or a 'Children in Focus' clinic had been attended by 19 July 1999). Of these initial pregnancies, there was a total of 14 676 fetuses, resulting in 14 062 live births and 13 988 children who were alive at 1 year of age. The aim of the study is to assess ways in which the environment (defined in its broadest sense) interacts with genetics to influence the health, development and well-being of the offspring.[20] To this end, data collection used a variety of methodologies including direct examination of the offspring, self-completion questionnaires administered to the parents, the children and their teachers, collection and assays of biological samples (including DNA), linkage to health and education records.[21 22] The study website contains details of all the data that are available through a fully searchable data dictionary and variable search tool: http://www.bristol.ac.uk/alspac/researchers/our-data/. ALSPAC has had its own ethics committee since its inception.[23]

### Outcomes

#### Identification of autism

In order to identify the children with autism we used the following sources: (1) a review of all children given a statement for special educational provision in the Avon area, to identify children diagnosed by age 11 as having special educational needs conforming to a diagnosis of autism using the International Classification of Diseases, 10th revision (ICD-10) criteria[24]; (2) the mother's answer to the question at age 9 'Have you ever been told that your child has autism, Asperger's syndrome or autistic spectrum disorder?'; (3) classification as Pervasive Development Disorder using questions from the Development and Well-Being Assessment (DAWBA) questionnaire at 91 months,[25] with the answers to the questionnaire classified by a child psychiatrist; (4) text responses to any question on diagnoses given to the child in questionnaires from 6 months to 11 years suggesting the child had been given an autism diagnosis; (5) ad hoc letters from parents to the Study Director. We considered that no one of these sources would be adequate, and so used all, and monitored the overlaps. In this way we identified 177 ALSPAC offspring with a probable diagnosis of autism—139 boys and 38 girls. Validation of the use of this group as autism

was shown by its strong relationship with a polygenic risk score for autism.[26]

## Autistic traits

We have used the upper 10th centile of the scores on the four independent trait predictors of autism identified previously as most predictive of autism in this cohort and described below. They included measures of social communication, coherent speech, sociability temperament and repetitive behaviour. Each had been shown to be an independent predictor of autism as identified using clinical records in this cohort.[27]

## Social communication trait

We used the 12-item Social and Communication Disorders Checklist (SCDC), developed by Skuse and colleagues.[28] They showed that the internal consistency was excellent (0.93) and the test-retest reliability was high (0.81). The method was developed on clinical samples, and when later used on the ALSPAC population at age 7.7 years the high end of the scale was shown to predict a variety of adverse outcomes, but was most specific for autism spectrum disorder.[29] Further research with ALSPAC data showed that the measure was reasonably stable over time.[30] For the present analysis we have used the prorated score, which was calculated when any items of the scale were missing a response, by using the average of the items that had been answered by the individual (2.7% of the population, almost all of whom had just one item missing). If all items were missing, the score was put to missing. The measure ranged from 0 to 24, and the higher the score the more impaired was the child's social cognition. The distribution was skewed with a long upper tail (12.8% had a score of over 6 and comprise the group with the highest autistic trait for these analyses).

## Coherence measure

At the children's age of 9, the mothers in the study completed a questionnaire which included seven of the nine scales of the first version of the Children's Communication Checklist (CCC).[31] This checklist was designed to assess aspects of communication that were not readily identified by conventional standardised tests, including aspects of speech and syntax as well as pragmatic aspects such as over-literal interpretation of stereotyped language. Although the CCC was initially designed to identify pragmatic difficulties, it has been shown to be good at discriminating a wide range of language and communication problems from typical development.[32] Analyses of traits predictive of autism in ALSPAC showed that the Coherence scale performed better than the other CCC scales[27] and consequently it is used here. The scale comprises eight items (eg, 'It is sometimes hard to make sense of what she is saying because it seems illogical or disconnected' and 'She has difficulty in telling a story or describing what she has done in a sequence of events'). The score ranged from 20 to 36, with higher scores indicating more typical behaviour. The score had

a skewed distribution. The lower tail used in this analysis comprised those children scoring <33 points (10.0% of the population).

## Abnormal and repetitive behaviour

This scale was developed from the answer to four questions in the questionnaire sent to the mother at 69 months; these were as follows: 'How often does he/she: (a) repeatedly rock his head or body for no reason; (b) have a tic or twitch; (c) have other unusual behaviour'; or (d) 'Does he/she stumble or get stuck on words, or repeat them many times? (eg, I I I I want a sweet)'? The responses to each question were coded as: often/always=3; sometimes=2; never=1 and summed. The resultant scale had a range from 4 to 12, with 22% scoring 5 and only 5.9% scoring more than 5. As it was impossible to approximate to a 10% cut-off, we used >5 as our group with the highest autistic trait.

## Sociability temperament

The questionnaire concerning the child which was sent to the study mothers when the child was 38 months of age included the 20 questions of the Emotionality, Activity and Sociability (EAS) Temperament scale[33]; this measured four traits— emotionality, activity, shyness, and sociability—each based on the answers to five questions. The range of the sociability sub-score was from 5 to 25 and the frequency distribution was approximately normal, a high score indicating a high level of sociability. The prorated scale was calculated for missing values as in the scales mentioned above. We then selected the lowest 11.4% of the children for our analyses (score <8) as being the nearest to 10% as the group with highest autistic trait.

## Maternal reports of ear and upper respiratory signs and symptoms

Questions regarding their child's ear and upper respiratory signs were sent to the mother by post when the child was aged 18, 30 and 42 months. Each questionnaire was structured with response boxes to be ticked. For the current study we used nine questions which were repeated at these three time points, and which described common ENT signs relating to hearing and the upper respiratory system. Details of the actual wording of the questions are shown in box 1. They cover aspects of breathing (mouth breathing, snoring, symptoms of sleep apnoea), ears (pulling or poking at ears, ears going red, pus discharging from ears), and subjective signs of hearing loss and listening difficulties (hearing worse during a cold, child rarely listens). Questions concerning whether or not the child had had earache were also asked, and the results are provided, but since the children were mostly too young to describe what was wrong, we have treated these data as less reliable than the other questions asked.

## Potential confounders

Adjustments were made for the following 10 potential confounders: preterm gestation (<37 weeks; 37+ weeks), sex, parity (defined as the number of previous pregnancies resulting in a live or stillbirth, 0 v 1+), breast feeding

**Box 1   Ear, nose and throat and hearing-related questions in the ALSPAC parental questionnaires**

Has anyone thought there may be a problem with his hearing?

Has your baby had earache?

Does s/he breathe through her mouth rather than through her nose?

Does s/he snore for more than a few minutes at a time?

When she is asleep, does s/he seem to stop breathing or hold her breath for several seconds at a time?*

Does she pull, scratch or poke at her ears?

Do his/her ears go red and look sore for a long time?

Has pus or a sticky mucus (not ear wax) ever leaked out of his/her ear?

During or after a cold, is his/her hearing worse than usual?

Generally, does your toddler listen to people or to things that happen nearby?

*Indicating sleep apnoea.

ALSPAC, Avon Longitudinal Study of Parents and Children.

(any vs none), maternal depression at 8 weeks post-delivery as assessed using the Edinburgh Postnatal Depression Score,[34] maternal educational achievements (5-point scale), maternal smoking at 18 weeks gestation (none vs any), maternal locus of control (using a shortened version of the adult version of the Nowicki-Strickland Internal-External locus of control scale as described elsewhere),[35] child's exposure to environmental tobacco smoke at 15 months measured as the length of time the child was in a room with others smoking at weekends and weekdays (any vs none), and attending a crèche or other type of daycare by 30 months (yes vs no). Table 1 shows frequencies of these confounders.

## Statistical analysis

Initial analysis used unadjusted comparisons between the different signs and symptoms and the five different binary outcomes (the most disadvantaged 10% of the four autistic traits and diagnosed autism). Logistic regression using STATA was then used to allow for the 10 potential confounders— with 120 logistic regressions being undertaken. In order to take account of multiple testing we determined how many associations would be expected to result in p<0.01 by chance for both the unadjusted and the adjusted associations. For each set of analyses just 1.2 of the 120 results were expected at p<0.01. Any excess in numbers of adjusted results at p<0.001 were assumed to be either indicating causality or the possibility of unadjusted confounding.

## Patient and public involvement

Patients and/or the public were not involved in the reporting or dissemination plans of this research. Before data collection (in the early 1990s), focus groups of pregnant women were involved in discussions on the content of questionnaires.

## RESULTS

### Unadjusted comparisons of ear and upper respiratory signs with autism outcomes

The responses to each question asked at 18, 30 and 42 months were compared between those scoring at the extreme ends of the autism trait scores, as well as for those diagnosed with autism in online supplemental tables 1–8. Of the 120 comparisons, 85 were at p<0.01 (1.2 expected) and 66 at p<0.0001 (0.001 expected).

Among the different ages tested, the most striking were at age 30 months (table 2) where it can be seen that those groups with the highest autistic traits reported more ear and upper respiratory signs. Autism itself was significantly associated with all signs except for symptoms of sleep apnoea. Conversely, the sociability trait was much less strongly associated with these ear and upper respiratory signs than any of the other three traits. Also note that pus leaking from the ears was only associated with abnormal scores on the speech coherence trait and with autism itself.

### Adjusted comparisons of ear and upper respiratory signs with autism outcomes

Results of adjustment for 10 factors (gestation, sex, parity, breast feeding, maternal depression, education, prenatal smoking, maternal locus of control, child exposure to environmental tobacco smoke, and starting crèche by age 30 months) are shown in table 3. It can be seen that of

**Table 1**   Frequencies of variables used as confounders

| Confounder | Number with data available | Frequency |
|---|---|---|
| Pre-term delivery | 12 935 | 5.3% <37 weeks |
| Sex of child | 14 676 | 51.8% male |
| Parity of mother | 12 787 | 48.2% primiparous |
| Whether breast fed at 7 days | 11 880 | 66.1% |
| Maternal depression at 2 months | 11 213 | 8.8% |
| Maternal education level (five levels) | 12 370 | From 20.8% lowest to 12.8% highest |
| Maternal smoking at 18 weeks gestation | 13 274 | 19.6% |
| Maternal locus of control | 12 471 | 45.8% external |
| Child's exposure to environmental tobacco smoke at 15 months | 11 073 | 42.0% exposed |
| Attended crèche or equivalent by 3 years | 10 038 | 36.3% exposed |

**Table 2** Summary of unadjusted results comparing % rates (n) of ear, nose and throat signs at 30 months of children who later demonstrated high levels of autistic traits and of children diagnosed with autism contrasted with controls

| Signs and symptoms | Group | Social communication % (n) | Speech coherence % (n) | Sociability % (n) | Repetitive behaviour % (n) | Autism % (n) |
|---|---|---|---|---|---|---|
| Mouth breathing | Index | 27.2 (232) | 26.5 (169) | 24.5 (226) | 24.1 (489) | 30.3 (42) |
| | Control | 19.1 (1119) | 19.8 (1145) | 20.8 (1524) | 19.0 (992) | 21.8 (1966) |
| | P value | *** | *** | * | *** | ** |
| Snoring | Index | 25.4 (221) | 22.0 (150) | 20.9 (196) | 20.2 (417) | 31.0 (41) |
| | Control | 16.0 (947) | 16.6 (967) | 17.4 (1283) | 16.1 (851) | 18.2 (1660) |
| | P value | *** | *** | ** | *** | *** |
| Sleep apnoea | Index | 18.1 (148) | 18.4 (110) | 15.6 (143) | 17.4 (339) | 16.8 (21) |
| | Control | 14.0 (802) | 13.9 (789) | 15.2 (1099) | 13.6 (705) | 15.5 (1387) |
| | P value | *** | *** | | *** | |
| Pulling at ears | Index | 8.2 (78) | 8.4 (60) | 6.8 (71) | 7.0 (157) | 16.2 (25) |
| | Control | 4.1 (267) | 4.2 (268) | 4.4 (353) | 3.7 (213) | 4.6 (463) |
| | P value | *** | *** | ** | *** | *** |
| Ears red and sore | Index | 21.0 (221) | 20.4 (139) | 18.0 (181) | 19.2 (418) | 26.5 (40) |
| | Control | 13.0 (820) | 13.5 (841) | 14.6 (1150) | 12.1 (680) | 15.0 (1458) |
| | P value | *** | *** | ** | *** | *** |
| Pus/mucus from ears | Index | 3.9 (37) | 5.1 (36) | 4.4 (46) | 4.0 (90) | 10.3 (16) |
| | Control | 3.2 (207) | 3.0 (189) | 3.3 (263) | 3.1 (177) | 6.6 (349) |
| | P value | | ** | | | *** |
| Hearing worse during cold | Index | 21.5 (148) | 25.8 (133) | 16.5 (127) | 20.1 (319) | 37.9 (44) |
| | Control | 14.1 (682) | 14.1 (673) | 15.6 (946) | 13.1 (574) | 15.2 (1136) |
| | P value | *** | *** | | *** | *** |
| Rarely listens | Index | 4.6 (44) | 6.4 (45) | 5.1 (53) | 4.1 (91) | 14.4 (22) |
| | Control | 1.6 (101) | 1.5 (95) | 1.7 (139) | 1.4 (80) | 2.1 (213) |
| | P value | *** | *** | *** | *** | *** |

*P<0.05; **p<0.01; ***p<0.001.

the 120 logistic regression results, 41 were associated at p<0.001 (0.0012 expected). The results are discussed in terms of the three categories of ear and upper respiratory signs below.

### Differences in breathing

The unadjusted comparisons showed that mouth breathing (all of the time or much of the time) was positively associated with autistic traits and autism at each of the three ages at which the question was asked, with p<0.05 for 14 of the 15 data points. Similarly, snoring at each of the three time points showed positive associations with the autism outcomes, all of the 15 comparisons being at p<0.05. However, for symptoms of sleep apnoea (often or sometimes), although there were always more affected children with the autistic outcome, only 10 of 15 associations showed differences at p<0.05 (online supplemental tables 1–3).

On adjustment (table 3) the three signs in this category showed different associations: social communication and coherence showed associations with all three signs, repetitive behaviour with two of the signs (mouth breathing and sleep apnoea), but sociability only with mouth breathing

and only at one time point (18 months). Although numbers with autism were small, the adjusted OR (aOR) was as high as 1.93 and 2.13 for mouth breathing and snoring, respectively.

### Signs of ear differences

The proportion of children who often pulled, scratched, or poked at their ears also showed positive unadjusted associations at p<0.05 in 11 of the 15 autism outcomes, and 14 of 15 associations for a history of the child's ears being 'red and sore looking'. 'Pus leaking from the ear (more than once)' was associated with a lower number of autistic outcomes (7 of 15), with particularly strong associations with speech coherence and autism (online supplemental tables 4–6).

After adjustment (table 3), pulling, scratching, or poking at ears was particularly strongly associated with autism, with aORs of 3.40 (95% CI 2.06 to 5.640) and 3.77 (95% CI 2.11 to 6.74) at 30 and 42 months, respectively. There were also strong associations for each autistic trait: the age at maximum odds varied from 30 months for social communication and repetitive behaviour, to 42 months for speech coherence and sociability. The adjusted

**Table 3** High levels of autistic traits or diagnosed autism by upper respiratory signs

| Outcome | Social communication<br>aOR (95% CI) | Speech coherence<br>aOR (95% CI) | Sociability<br>aOR (95% CI) | Repetitive behaviour<br>aOR (95% CI) | Autism<br>aOR (95% CI) |
|---|---|---|---|---|---|
| Mouth breathing | | | | | |
| 18 months | 1.12 (0.90 to 1.39) | 1.51 (1.21 to 1.81)*** | 1.34 (1.07 to 1.67)** | 1.36 (1.14 to 1.62)*** | 2.13 (1.32 to 3.44)** |
| 30 months | 1.41 (1.20 to 1.67)*** | 1.31 (1.10 to 1.55)** | 1.11 (0.93 to 1.33) | 1.24 (1.09 to 1.43)** | 1.62 (1.07 to 2.43)* |
| 42 months | 1.43 (1.22 to 1.67)*** | 1.32 (1.12 to 1.57)*** | 1.09 (0.92 to 1.30) | 1.31 (1.15 to 1.49)*** | 1.40 (0.93 to 2.11) |
| Snoring | | | | | |
| 18 months | 1.72 (1.23 to 2.41)** | 1.92 (1.35 to 2.72)*** | 1.12 (0.76 to 1.64) | 1.31 (0.97 to 1.77) | 1.38 (0.55 to 3.43) |
| 30 months | 1.54 (1.18 to 2.03)** | 1.32 (0.98 to 1.78) | 1.13 (0.83 to 1.53) | 1.21 (0.95 to 1.52) | 1.54 (0.77 to 3.11) |
| 42 months | 1.47 (1.15 to 1.88)** | 1.33 (1.02 to 1.73)* | 1.00 (0.75 to 1.33) | 1.20 (0.97 to 1.48) | 1.93 (1.09 to 3.41)* |
| Sleep apnoea | | | | | |
| 18 months | 1.54 (1.20 to 1.96)*** | 1.23 (0.93 to 1.62) | 0.89 (0.66 to 1.19) | 1.33 (1.08 to 1.63)** | 1.14 (0.57 to 2.28) |
| 30 months | 1.24 (1.02 to 1.50)* | 1.30 (1.06 to 1.59)* | 0.91 (0.73 to 1.13) | 1.17 (1.00 to 1.37)* | 0.93 (0.53 to 1.63) |
| 42 months | 1.28 (1.06 to 1.55)** | 1.33 (1.09 to 1.62)** | 0.99 (0.80 to 1.23) | 1.20 (1.03 to 1.41)* | 1.35 (0.84 to 2.17) |
| Pulls/pokes at ears | | | | | |
| 18 months | 1.42 (1.13 to 1.79)** | 1.11 (0.86 to 1.44) | 1.24 (0.96 to 1.59) | 1.42 (1.17 to 1.73)*** | 1.34 (0.75 to 2.41) |
| 30 months | 1.57 (1.20 to 2.07)*** | 1.52 (1.14 to 2.03)** | 1.48 (1.10 to 1.99)** | 1.78 (1.41 to 2.25)*** | 3.40 (2.06 to 5.64)*** |
| 42 months | 1.33 (0.93 to 1.90) | 1.92 (1.38 to 2.69)*** | 1.74 (1.24 to 2.46)*** | 1.52 (1.14 to 2.03)** | 3.77 (2.11 to 6.74)*** |
| Ears turn red | | | | | |
| 18 months | 1.19 (1.01 to 1.40)* | 1.09 (0.92 to 1.29) | 1.13 (0.95 to 1.34) | 1.46 (1.28 to 1.66)*** | 1.10 (0.72 to 1.69) |
| 30 months | 1.35 (1.13 to 1.61)*** | 1.17 (0.96 to 1.41) | 1.20 (0.99 to 1.45) | 1.54 (1.33 to 1.79)*** | 2.14 (1.44 to 3.17)*** |
| 42 months | 1.27 (1.05 to 1.53)* | 1.21 (0.99 to 1.47) | 1.05 (0.86 to 1.30) | 1.33 (1.14 to 1.56)*** | 1.52 (0.98 to 2.37) |
| Pus/mucus from ears | | | | | |
| 18 months | 0.98 (0.65 to 1.48) | 1.27 (0.86 to 1.90) | 1.26 (0.86 to 1.85) | 1.14 (0.83 to 1.57) | 1.87 (0.85 to 4.08) |
| 30 months | 1.12 (0.78 to 1.60) | 1.32 (0.92 to 1.89) | 1.19 (0.83 to 1.71) | 1.33 (1.00 to 1.75)* | 3.29 (1.85 to 5.86)*** |
| 42 months | 1.31 (1.01 to 1.72)* | 1.23 (0.93 to 1.64) | 1.22 (0.91 to 1.65) | 1.31 (1.05 to 1.64)* | 2.85 (1.72 to 4.72)*** |
| Hearing worse during a cold | | | | | |
| 18 months | 1.22 (0.88 to 1.71) | 1.53 (1.10 to 2.14)* | 1.22 (0.86 to 1.73) | 1.25 (0.95 to 1.64) | 0.82 (0.33 to 2.06) |
| 30 months | 1.33 (1.10 to 1.62)** | 1.35 (1.11 to 1.65)** | 1.22 (1.00 to 1.49) | 1.40 (1.20 to 1.64)*** | 2.18 (1.43 to 3.31)*** |
| 42 months | 1.47 (1.20 to 1.79)*** | 1.73 (1.41 to 2.12)*** | 1.10 (0.88 to 1.37) | 1.59 (1.35 to 1.88)*** | 3.28 (2.15 to 5.00)*** |
| Rarely listens to nearby sounds | | | | | |
| 18 months | 2.40 (1.61 to 3.58)*** | 1.58 (1.00 to 2.51) | 2.07 (1.40 to 3.06)*** | 1.89 (1.32 to 2.72)*** | 2.26 (0.97 to 5.27) |
| 30 months | 2.56 (1.76 to 3.73)*** | 2.69 (1.82 to 3.97)*** | 2.87 (2.01 to 4.10)*** | 2.57 (1.84 to 3.59)*** | 6.44 (3.68 to 11.28)*** |
| 42 months | 2.64 (1.96 to 3.57)*** | 2.76 (2.01 to 3.79)*** | 2.31 (1.70 to 3.15)*** | 2.11 (1.60 to 2.79)*** | 6.60 (4.04 to 10.77)*** |

OR (95% CI) adjusted for gestation, sex, parity, breast feeding, maternal depression, maternal education, prenatal smoking, locus of control, child exposure to environmental tobacco smoke and starting crèche (see main text for details).
*P<0.05; **p<0.01; ***p<0.001
aOR, adjusted OR;

odds were less pronounced for children whose ears were reported to have gone red for prolonged periods of time and were only associated with two traits (social communication and repetitive behaviour). For autism the association was strongest at 30 months with an aOR of 2.14 (95% CI 1.44 to 3.17). For pus from the ears, there were only mild associations with the autistic traits, but strong associations with autism, particularly at 30 months with an aOR of 3.29 (95% CI 1.85 to 5.86).

### Signs of listening and hearing difficulties
On unadjusted analyses, reports of the child's hearing being worse during or after a cold were associated with autism and for three of the four autistic traits (sociability being the exception). The differences of the four outcomes were apparent at each of the three ages at p<0.0001 (online supplemental table 7). The unadjusted pattern was different in relation to the question concerning whether the child listens to people or to things that happen nearby. Here all comparisons were at p<0.05, including the poor sociability outcome (online supplemental tables 7–8).

On adjustment, the results are similar to the unadjusted associations (table 3), with the aORs being particularly high for failure to react to noise nearby at 30 months of

age for children with autism (aOR 6.44, 95% CI 3.68 to 11.28, p<0.0001).

## DISCUSSION

There is increasing interest worldwide in autism, and much research has been undertaken to understand the neurocognitive and genetic differences.[36] In this set of analyses we have shown that common indicators of ear health and upper respiratory compromise are more frequent in children subsequently identified with autism as well as in children who have high levels of several autistic traits. These associations may be important because (1) these ear and respiratory signs may be early markers of increased risk of autism, (2) they may inform the origins of autism, or (3) they may highlight co-occurring conditions that if treated may lead to a better quality of life for children with autism.

Tto our knowledge, no birth cohort studies have investigated prospectively the common signs that might indicate upper respiratory disadvantage such as snoring, mouth breathing or sleep apnoea as precursors of autism. Sleep problems are commonly reported by parents of children with autism, including problems due to sleep-related breathing disorders, such as snoring or apnoea.[37] In this study we found that mouth breathing in the first 3 years of life was associated with autism, but not symptoms of sleep apnoea, or early snoring except at 42 months. However, both snoring and symptoms of sleep apnoea in the early years were linked to the autistic traits of social communication and speech coherence, across most of the time points; sleep apnoea was also linked to repetitive behaviour. Studies examining whether sleep-related breathing disorders are more commonly reported in autism than in typically developing children are generally case–control studies of children with a confirmed diagnosis of autism. Results generally show more sleep disordered breathing in children with autism,[10 11 38–40] although Alfonso-Alfonso et al[41] found no difference, and Malow and colleagues[42] carried out polysomnography in children with autism and typically developing controls and found no physiological evidence of sleep apnoea in the autism group. Our findings based on a large sample size may indicate that children who develop autistic traits have a slightly disordered upper respiratory structure or physiology, such as obstructed airways from enlarged adenoids. For example, snoring occurs during sleep when the upper respiratory system relaxes and various organs in the airways vibrate, and breathing through the mouth rather than the nose is an indicator of obstructed airways, as is breath holding or stopping breathing during sleep.

Similarly, subtle signs of possible middle ear disease, such as pulling or poking at the ears, the ears going red for a prolonged period of time, or pus or sticky mucus discharging from the ears, appear not to have been considered before in birth cohort studies. We found that these signs were closely associated with autistic traits and with a diagnosis of autism, especially if the signs were present

at around 30 months of age. There were also strong relationships between ear discharge and autism. Ear infections have been linked to autism based on medical attendance for otitis media in health registry studies,[4 7 8] through prospective parental report of ear infections in a birth cohort study,[19] and retrospective parental report in a case–control study.[14] Interestingly, a study based on the Danish National Birth Cohort found no association between parental report of ear infections and autism, but did find an association when hospital contact for ear infections was examined as the exposure[43]; the authors suggest the hospital record data could be affected by both detection and selection bias which is less likely to influence studies involving prospective parental reporting.

In order to determine whether the child's hearing was affected in the absence of direct hearing tests, we assessed the responses to questions concerning whether the child's hearing was thought by the mother to be worse during a cold, and whether she reported that the child rarely listened to people or things that happened nearby. Both characteristics were associated with autism and autistic traits, particularly at 30 months. Although unresponsive behaviour is a feature of the social communication autistic trait, the associations with listening difficulties accompanying a cold implies the presence of a fluctuating conductive hearing loss rather than inattention. This finding is consistent with case–control studies that have shown higher levels of abnormal tympanograms (middle ear function tests indicating the presence of otitis media) in autism.[15 44 45] There is limited evidence, however, that older children with autism have peripheral hearing loss, with relevant studies systematically reviewed in 2014.[46]

This study adds to the evidence that, compared with a typical population of the same age, early ear and upper respiratory symptoms are more common in those subsequently diagnosed with autism or with extreme levels of autistic traits. It is not possible to determine whether these ENT conditions have a causal role in the development of autistic traits or are related to an unmeasured factor. One possibility, for example, could be the consequence of the increased prevalence of minor physical anomalies in individuals with autism,[47] including anatomical differences in the structure and/or positioning of the ear,[14 48] with such differences in ear morphology increasing the risk of ENT conditions.

Ongoing OME during early childhood leads to impairment in auditory processing, although this improves as the otitis media clears.[49] Atypical sensory processing is a common presentation in children with autism,[50 51] but also the prevalence of autism is higher in deaf children than hearing children and similarly higher in children with visual impairment.[52] This suggests that in children with an existing susceptibility to development of autism, disrupted auditory processing as a result of otitis media may interact with existing deficiencies in sensory processing and integration, impacting on the development of autistic traits.[50]

Aside from suggestions as to mechanisms, it is clear from this study of prospectively collected information that children who later develop social communication difficulties are more likely to have early middle ear disease and ENT conditions, and are therefore more at risk of communication difficulties from hearing loss, although temporary. Early detection and intervention of ENT conditions in children with autism is thus likely to be beneficial.

## Strengths and limitations

An important strength of this study lies in the fact that the population is geographically defined and includes the great majority (~80%) of the eligible population. The data are possibly unique in documenting common early ear and upper respiratory signs and symptoms of the children who subsequently develop extreme levels of autistic traits and/or a diagnosis of autism. The mothers answered structured questions at distinct time points with no idea as to whether the signs they were describing were linked to outcomes such as autism. Thus, there were no discernible biases in ascertainment of these signs, or of the later identification of the autistic traits.

The limitations, as in all longitudinal studies, lie in the loss of children to later follow-up.[21] Unfortunately, at the time the children were born there were very few families of ethnic minorities resident in the area (~6%). Consequently, our results cannot be extrapolated to cover non-white populations in general. Another limitation concerns the fact that, to our knowledge, the questions used to record ear and upper respiratory symptoms have not been validated. However, they asked about objective rather than subjective signs and symptoms, and parents gave similar results over the childhood years. A further limitation concerns the fact that the study children were not examined consistently to determine a diagnosis of autism; rather, a strategy to assess the probability of a diagnosis using a variety of different sources was used. The validity of this approach was shown since the group of children identified in this way demonstrated a positive correlation with a polygenic risk score for autism.

It is always possible that the statistical analyses did not allow for all the appropriate confounders, and further datasets are needed to assess the validity of our results.

## CONCLUSION

Data collected longitudinally from 18 to 42 months have demonstrated close associations of common early ear and upper respiratory markers with a later diagnosis of autism as well as with high levels of autistic traits. These markers included snoring and mouth breathing as well as a reduction in hearing ability during a cold, and signs of OME. The data indicate that the strongest associations occurred at the age of 30 months. These results may indicate disruptions in early auditory perception and processing in the early stages of autism.

**Acknowledgements** We are extremely grateful to all the families who took part in this study, the midwives for their help in recruiting them, and the whole ALSPAC team, which includes interviewers, computer and laboratory technicians, clerical workers, research scientists, volunteers, managers, receptionists, and nurses.

**Contributors** Conception and design of study: AH and JG. Acquisition of funding: JG and YIC. Preparation of study data and analyses: AH and SG. Statistical analysis quality checking: JG and AH. Original draft: AH and JG. JG, AH, RM, DR, YIC contributed to interpretation of the findings. JG, AH, RM, DR, YIC, SG contributed to the critical review and editing of the manuscript and approved the final manuscript. All authors are guarantors for this work. The corresponding author attests that all listed authors meet authorship criteria and that no others meeting the criteria have been omitted.

**Funding** This publication was made possible through the support of grants from the Medical Research Council (G0400085) and the John Templeton Foundation (ref no. 61917). The UK Medical Research Council and Wellcome Trust (Grant ref: 217065/Z/19/Z) and the University of Bristol (grant ref: N/A) currently provide core support for ALSPAC. This publication is the work of the authors who will serve as guarantors for the contents of this paper. A comprehensive list of grants funding is available on the ALSPAC website (http://www.bristol.ac.uk/alspac/external/documents/grant-acknowledgements.pdf). The opinions expressed in this publication are those of the author(s) and do not necessarily reflect the views of the John Templeton Foundation.

**Competing interests** JG and RM are retired; YIC and SG are funded by the John Templeton Foundation (61917). The authors declare no financial relationships with any organisations that might have an interest in the submitted work in the previous 3 years and no other relationships or activities that could appear to have influenced the submitted work.

**Patient and public involvement** Patients and/or the public were involved in the design, or conduct, or reporting, or dissemination plans of this research. Refer to the Methods section for further details.

**Patient consent for publication** Not applicable.

**Ethics approval** This study involves human participants and ethical approval for the study was obtained from the ALSPAC Ethics and Law Committee (ALEC; IRB00003312) and the Local Research Ethics Committees. Detailed information on the ways in which confidentiality of the cohort is maintained may be found on the study website: http://www.bristol.ac.uk/alspac/researchers/research-ethics/ All methods were performed in accordance with the relevant guidelines and regulations. Implied informed consent for the use of data collected via questionnaires and clinics was obtained from participants following the recommendations of the ALSPAC Ethics and Law Committee at the time.

**Provenance and peer review** Not commissioned; externally peer reviewed.

**Data availability statement** Data may be obtained from a third party and are not publicly available. ALSPAC data are available to researchers for particular projects, provided no attempt is made to reveal the identities of the subjects. Guidelines for access are found on the ALSPAC website: www.bristol.ac.uk/alspac/researchers.

**ORCID iDs**
Amanda Hall http://orcid.org/0000-0001-8520-6005
Jean Golding http://orcid.org/0000-0003-2826-3307

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
