## [Reviewer comments · BMJ Open]

ARTICLE DETAILS

TITLE (PROVISIONAL)	Associations between autistic traits and early ear and upper respiratory signs: a prospective observational study of the ALSPAC geographically defined childhood population
AUTHORS	Hall, Amanda; Maw, Richard; Iles-Caven, Yasmin; Gregory, Steven; Rai, Dheeraj; Golding, Jean

VERSION 1 – REVIEW

REVIEWER	Norazlin Kamal Nor National University of Malaysia, Dept of Paediatrics
REVIEW RETURNED	20-Sep-2022

GENERAL COMMENTS	One clear disadvantage- which is mentioned upfront- is that not all children in the study were assessed for a diagnosis of autism. I wonder if was there an attempt to screen for neurodevelopmental conditions or ASD risk for example with M-CHAT-R/F or a similarly recognized screening tool for ASD? In terms of ASD diagnosis, I noticed the diagnosis of ASD was made by ICD-10, as well as a variety of other means to potentially identify children with ASD. I do think that having a standardized measure such as an ADOS assessment would make the ASD diagnosis more reliable and robust. I wonder how the ENT and hearing-related questions were developed? Were the questions from a validated study, and how well did they measure what they purport to measure? Overall I found this study to be compelling and well thought-out. Challenges were mentioned outright and conclusions reached were measured and reasonable. I think this is an important research output that should published. Congratulations on your admirable work.
---

REVIEWER	Shui Yin Lo Quantum Health Research Institute
REVIEW RETURNED	12-Jan-2023

GENERAL COMMENTS	I suggest the paper should include in the introduction reference to the following paper and book: 1. Diagnosis, Treatment and Prevention of Autism via Meridian Theory, The American Journal of Chinese Medicine, Vol. 40, No. 1, 39–56 © 2012 World Scientific Publishing, by Shui Yin Lo Quantum Health Research Institute Pasadena, CA, USA American University of Complementary Medicine Beverly Hills, CA, USA Company
---

	2. "Autism and Stable Water Clusters, Physics and Health: A Picture book, by Shui Yin Lo, Publisher Quantum Health Research Institute 2013", available at amazon.com
--	--

VERSION 1 – AUTHOR RESPONSE

1. In response to Reviewer 1:

She wondered if was there an attempt to screen for neurodevelopmental conditions or ASD risk for example with M-CHAT-R/F or a similarly recognized screening tool for ASD? Alas, the questionnaires were designed before there were validated tools available to identify cases of autism. This is why we have used the traits which were valid predictors of ASD in our study as well as using multiple sources to define the cases. As we said in the limitations: ‘A further limitation concerns the fact that the study children were not examined consistently to determine a diagnosis of autism. Rather a strategy to assess the probability of a diagnosis using a variety of different sources was used. The validity of this approach was shown since the group of children identified in this way demonstrates a positive correlation with a polygenic risk score for autism’ (JAMA Psychiatry 2018;75:835-43).

b) She wondered how the ENT and hearing-related questions were developed, whether the questions were from a validated study, and how well they measured what they purport to measure. The respiratory questions used in this paper were chosen with advice from a number of experts in the respiratory and ENT fields; we also took account of questions that had been asked in previous longitudinal studies. To our knowledge, none of the questions have been validated; however, the questions do mainly relate to objective signs such as snoring, pus discharge from ears, poking or pulling at ears, and ears going red.

We are really pleased that she found this study to be ‘compelling and well thought-out’ and would like to thank her.

Reviewer: 2

We are grateful to Dr Lo for alerting us to his paper, but since it only refers to a pilot study we did not think it was appropriate for this paper.

VERSION 2 – REVIEW

REVIEWER	Norazlin Kamal Nor National University of Malaysia, Dept of Paediatrics
REVIEW RETURNED	01-Feb-2023

GENERAL COMMENTS	Congratulations on a commendable study and an engaging article. I believe this article will add to the knowledge in this field and I have recommended for this article to be published.
---